# The Arabic Version of the Faces Pain Scale-Revised: Cultural Adaptation, Validity, and Reliability Properties When Used with Children and Adolescents

**DOI:** 10.3390/children8121184

**Published:** 2021-12-15

**Authors:** Jessica Finianos, Elisabet Sánchez-Rodríguez, Jordi Miró

**Affiliations:** 1Unit for the Study and Treatment of Pain—ALGOS, Department of Psychology, Research Center for Behavior Assessment (CRAMC), Universitat Rovira i Virgili, 43007 Tarragona, Catalonia, Spain; jessica.finianos@estudiants.urv.cat (J.F.); elisabet.sanchez@urv.cat (E.S.-R.); 2Institut d’Investigació Sanitària Pere Virgili, 43007 Tarragona, Catalonia, Spain

**Keywords:** Faces Pain Scale-Revised, Arabic, validity, reliability, pain assessment, pain intensity, children

## Abstract

The Faces Pain Scale-Revised (FPS-R) is widely used to assess pain intensity in young people. The aims of this research were to study the convergent and discriminant validity and reliability properties of a culturally adapted version of the FPS-R for its use with Arabic-speaking individuals. The sample consisted of 292 students living in Lebanon. They were interviewed online, asked to imagine themselves in one of two given situations based on their age (8–12 and 13–18 years old), and then asked rate the intensity of pain they would experience using the FPS-R-Arabic and a Numerical Rating Scale (NRS-11-Arabic). They were also asked to respond to the Pain Catastrophizing Scale (PCS-C-Arabic). Two weeks later, participants were asked to repeat the same procedure. The data showed strong associations between the scores of the FPS-R-Arabic and NRS-11-Arabic (r = 0.72; *p* < 0.001), which were higher than the associations of the scores of the FPS-Arabic with the PCS-C-Arabic scores (z = 7.36, *p* < 0.001). The associations between the FPS-R-Arabic scores on the two measurements were also strong (r = 0.76; *p* < 0.001). The findings support the convergent and discriminant validity and reliability of the FPS-R-Arabic scores when used to measure pain intensity in young people aged 8 to 18 years old.

## 1. Introduction

Pain is a common experience in children and adolescents [1,2,3,4]; but, even so, assessing pain in children is still a challenge [5,6,7]. Research has shown that self-reporting is the best and primary source of information for assessing pain intensity in children [8,9,10]. Various self-report questionnaires exist for measuring and assessing pain intensity, all with positive and negative characteristics [9,11,12,13,14].

Faces are commonly used in self-report pain intensity questionnaires, especially with younger children because they are appealing, simple, and easy to use [15]. Of the different faces scales, the revised version of the Faces Pain Scale [16,17] is one of the most used for measuring and assessing pain intensity [18,19,20]. The Faces Pain Scale-Revised (FPS-R) has six facial expressions and has two main advantages over other faces self-report pain intensity measures: namely, pain intensity scores can be matched with the common 0–10 metric used by most scales, like the numerical rating scale [17], and the faces are depicted without smiles or tears, thus avoiding the confusion between pain intensity and distress [19,21,22,23].

Currently, as well as its original English version, the FPS-R’s instructions are available in 69 languages (https://www.iasp-pain.org/resources/faces-pain-scale-revised/#download; last accessed 4 November 2021). However, not all these versions have been thoroughly studied, and there is no information about their psychometric characteristics [8,24,25,26].

The FPS-R has been translated into Arabic, but the psychometric properties of the translated instrument have not been studied. Although the FPS-R is very simple and easy to use [8,15,16] and is widely used with children and adolescents [13,19,27,28,29], it should not be assumed that the strong psychometric properties reported in studies of samples from countries with different languages and cultures remain the same when it is used with Arabic-speaking samples [8,30,31,32,33]. Thus, before it is recommended for general use and implementation, its psychometric properties should be subject to rigorous research. Indeed, research on the Arabic version of the FPS-R is important, as it could help improve how pain intensity is measured and assessed in Arabic-speaking children around the world [34,35], most of whom are monolingual or report difficulties when using a second language [36,37]. It would also help facilitate transcultural studies about the expression of pain.

The objectives of this research were to study the construct validity (i.e., convergent and discriminant validity) and reliability (i.e., test-retest reliability) of a culturally adapted version of the FPS-R for use with Arabic-speaking children and adolescents. If the scores of the FPS-R-Arabic were to be valid, we hypothesized that they would show a strong and positive statistically significant association with pain-intensity scores of the Arabic version of the Numerical Rating Scale (NRS-11-Arabic). We also hypothesized that the magnitude of the association between these two scores would be greater than between the FPS-R-Arabic and the Arabic version of Pain Catastrophizing Scale [38], a theoretically different scale to pain intensity, supporting discriminant validity. Finally, we hypothesized a strong and positive statistically significant association between scores of the FPS-R-Arabic at two different times, which would support its test-retest reliability properties.

## 2. Materials and Methods

### 2.1. Participants

A total of 292 children and adolescents living in Cada Zgharta and Beirut (Lebanon) participated in this study. This convenience sample was recruited via flyers shared on social media and a snowball strategy. To take part, eligible participants had to be between 8 and 18 years old and fluent in Arabic. Interested individuals were excluded if they had cognitive disabilities or failed to provide an informed consent form signed by their parents or their assent.

### 2.2. Measures

#### 2.2.1. Sociodemographic Information

Children and adolescents participating in the study were asked to provide information about their gender, age, and school grade.

#### 2.2.2. Pain

Participants were asked to report their pain intensity using both the Arabic versions of the Faces Pain Scale-Revised (FPS-R-Arabic) and the 0–10 Numerical Rating Scale (NRS-11-Arabic).

The NRS-11 is a one item questionnaire that assesses pain intensity by asking the child to estimate his or her pain using numbers from 0, referring to “no pain”, to 10, referring to “very much pain”. Pain intensity reports of the NRS-11 have been shown to be valid and reliable when used with pediatric populations [39,40,41], including Arabic-speaking samples [42].

In this study, we had to adapt the FPS-R for use with Arabic-speaking children. In the original English version, the FPS-R has six faces showing ascending pain intensity (from left to right). The first face (the leftmost) is described as showing “no pain”, and the sixth (the rightmost) is described as showing “very much pain”. However, in order to use a culturally sensitive form of the FPS-R, we had to reverse the position of the faces because Arabic individuals read from right to left. Therefore, in the Arabic version that we have used in this study, the faces showed an ascending level of pain from right to left, so the instructions also had to be slightly changed from the originals to reflect the different order of the faces.

#### 2.2.3. Pain Catastrophizing

In addition to pain intensity, participants were asked to respond to the 13-item Arabic version of the Pain Catastrophizing Scale-Child (PCS-C; 38) to measure catastrophic thinking about pain, in which respondents indicate how frequently they have catastrophic beliefs when in pain on a scale from 0 (“not at all”) to 4 (“extremely”). Scores range from 0 to 52, and the higher the score, the more the respondent catastrophizes about pain. The PCS-C assesses three pain catastrophizing domains: rumination (e.g., “I cannot keep it out of my mind”), magnification (e.g., “I am afraid that pain will get worse”), and helplessness (e.g., “There is nothing I can do to reduce pain”). However, in this study, we only used the total score. The PCS-C scores have been shown to have sound psychometric properties in different languages and in different samples of children and adolescents [42,43,44]. Cronbach’s alpha of the Arabic version of the PCS-C used in this study was excellent (α = 0.91).

### 2.3. Procedure

This study was approved by the Ethics Committee of the Universitat Rovira i Virgili (CEIPSA-2020-TD-0002) and conducted fully online in April and May of 2021. Recruitment flyers were posted and shared on social media (via Whatsapp, Instagram, and Facebook) seeking parents of children and adolescents aged between 8 and 18 years old living in Lebanon. The parents who showed an interest were asked permission for their child to participate. Six of the parents approached refused the request. The parents who agreed to proceed with the study were called to an online meeting.

During the first online meeting, the researcher met each participant and her or his mother or father, introduced herself, and explained the study procedure. Parents of participants younger than 13 years old were invited to stay in the meeting to provide technical help if needed as long as they remained silent. However, no parents had to intervene.

After obtaining the parents’ consent and participant’s assent, the researcher asked each participant to report the following information: gender, age, school grade, and pain if any. Then, the researcher sent the participant a link to the online survey and asked him/her to respond to the questions on pain intensity in her presence. When clicking on the link, participants found five sequential yet different pages. However, they could only move forward to the next pages of the survey by approving the first page, the informed consent, the demographic questions, the FPS-R-Arabic, the NRS-11-Arabic, and the PCS-C-Arabic, each following on a different page. Once the survey had been filled in and submitted, a thank-you note appeared on the participant’s screen, and a report with all the answers given was sent by email to the researcher under a randomly assigned code for each participant that was secured on a two-password-protected laptop.

Two different scenarios were used to elicit pain intensity reports with the FPS-R-Arabic and the NRS-11-Arabic. We followed a procedure used with similar objectives [8] and asked the participants to imagine themselves in one potentially painful situation or another depending on their age. Children of 8–12 years old were asked to imagine that they fell over and scraped their knees, whereas adolescents of 13–18 years old were asked to imagine that they burned their hand. Then, the researcher read the instructions for the FPS-R-Arabic and NRS-11-Arabic and asked the participant to report the pain intensity on the online survey, one after the other. Subsequently, participants were asked to respond to the PCS-C-Arabic before a second meeting was scheduled for two weeks later, when they would report their pain intensity with the FPS-R-Arabic and the NRS-11-Arabic using the same scenario and procedure of the first meeting.

### 2.4. Data Analysis

To describe the study sample, we computed the descriptive statistics of the demographic variables (percentages, means, and standard deviations). Then, to study the convergent validity of the FPS-R-Arabic scores, we computed a Pearson correlation coefficient between the ratings on the FPS-R-Arabic and the NRS-11-Arabic. Next, we conducted a Steiger’s *z*-test [45] to evaluate the discriminant validity of the scores of the FPS-R-Arabic by comparing the magnitude of the correlation between the ratings on the FPS-R-Arabic and the NRS-11-Arabic with the magnitude of the correlation between the FPS-R-Arabic and the PCS-C-Arabic. Finally, to study reliability, we computed a Pearson correlation coefficient between ratings on the FPS-R-Arabic at time 1 and time 2. To study the planned associations, we examined the group as a whole and also in two age groups (the 8–12-year-olds and the 13–18-year-olds) since we used two different scenarios. All analyses were conducted using the Statistical Package for Social Sciences for Windows version 27.0 (SPSS Inc., Chicago, IL, USA).

## 3. Results

### 3.1. Participants

A sample of 292 children and adolescents participated in this study. The average age was 13.11 years (SD = 2.73), and there were slightly more females (51%).

### 3.2. Validity

#### 3.2.1. Convergent Validity

Table 1 provides the Pearson correlations between the scores of the FPS-R-Arabic and those of the NRS-11-Arabic for the whole sample and both age groups. The correlations are strong and statistically significant and support the convergent validity of the FPS-R-Arabic scores.

#### 3.2.2. Discriminant Validity

The magnitude of the correlation between the scores on the FPS-R-Arabic and the scores on the NRS-11-Arabic was significantly greater than the magnitude of the correlation between the scores on the FPS-R-Arabic and the scores on the PCS-C-Arabic for the whole sample (z = 7.36, *p* < 0.001), for the 8–12 year-olds (z = 5.07, *p* < 0.001), and for the 13–18 year-olds (z = 4.91, *p* < 0.001).

### 3.3. Reliability

The test-retest reliability coefficients for the FPS-R-Arabic scores were good (r = 0.76 for the whole sample, 0.74 for the 8–12-year-olds, and 0.76 for the 13–18-year-olds).

## 4. Discussion

The purpose of this study was to evaluate the construct validity (i.e., convergent and discriminant) and reliability of the FPS-R-Arabic when used with children and adolescents to assess pain intensity. The data supported the hypothesis. That is, they showed a strong statistically significant positive association between the scores of the FPS-R-Arabic and the NRS-11-Arabic. Moreover, the magnitude of the associations between the FPS-R-Arabic and NRS-11-Arabic scores was greater than those between the FPS-R-Arabic and the PCS-C-Arabic scores, thus showing that the scores had good convergent and discriminant validity. The data also showed a strong statistically significant positive association between the scores of the FPS-R-Arabic at the two times of testing, thus showing that the scores had a good test-retest reliability.

The findings of this study extend those from previous research on the use of the FPS-R to measure pain intensity in children and adolescents between 8–18 years old [8,15,17,20] and is the first in evaluate its use in Arabic-speaking populations.

This study has limitations that should be taken into consideration when interpreting the findings. First, the study used a convenience sample of children and adolescents living in two Lebanese regions that may or may not be representative of the Arabic-speaking population. Therefore, studies need to be made with other samples from other Arabic-speaking countries to replicate the findings and determine which ones are valid. Nevertheless, the findings of this research do not differ from those of other studies using different versions of the FPS-R in samples with different languages [8,24,26,46,47]. Second, the nature of the procedure that requires imagination might have affected the results since some of the children might not have experienced burning their hand; yet, the procedure was used as it had been in several previous studies [48,49]. Third, the presentation of the pain intensity scales was not randomized, making it unclear whether the order influenced the ratings’ report. Yet again, however, previous similar studies found that the order in which the scales are presented does not make a difference [40,50,51].

Future studies should examine other psychometric properties (e.g., feasibility, sensitivity to change over time) that were not evaluated in this research and evaluate the influence of previous pain experiences on the interpretation of the top anchor of the questionnaire. Importantly, studies with samples of children with acute and chronic pain are also needed. Moreover, research with different age groups is needed to determine the minimum age at which the FPS-R-Arabic can be used, validly and reliably, in this population [8,16,17]. Moreover, future studies should also compare the FPS-R-Arabic scores with those from other common pain intensity scales, like the Visual Analogue Scale, to evaluate their agreement and facilitate their interchangeable use according to the child’s age and preferences.

Despite the study’s limitations, and in agreement with previous findings, the FPS-R-Arabic scores have shown strong validity and reliability properties when used with children and adolescents. The findings, if shown to be valid in future studies, would support the use of the FPS-R-Arabic to measure and assess pain intensity in young people. Having easy-to-administer, culturally adapted intensity scales with sound psychometric properties tested in the native language of the children with whom it will be used may encourage pain to be assessed, and this, in turn, may improve pain management and result in positive clinical outcomes.

## Figures and Tables

**Table 1 children-08-01184-t001:** Correlations between the different measures used in the study.

Whole Sample (*N* = 292)		
	FPS-R-Arabic	PCS-C-Arabic
NRS-11-Arabic	0.72 *	0.44 *
PCS-C-Arabic	0.38 *	
8–12 years old (*N* = 137)		
	FPS-R-Arabic	PCS-C-Arabic
NRS-11-Arabic	0.73 *	0.47 *
PCS-C-Arabic	0.40 *	
13–18 years old (*N* = 155)		
	FPS-R-Arabic	PCS-C-Arabic
NRS-11-Arabic	0.71 *	0.42 *
PCS-C-Arabic	0.39 *	

Note: FPS-R-Arabic, Arabic version of the Faces Pain Scale-Revised; PCS-C-Arabic, pediatric form of the Arabic version of the Pain Catastrophizing Scale; NRS-11-Arabic, Arabic version of the Numerical Rating Scale-11. * *p* < 0.001.

## Data Availability

The data presented in this study are available on request from the corresponding author. The data are not publicly available due to ethical restrictions and privacy.

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
