# Peer review of "The Arabic Version of the Faces Pain Scale-Revised: Cultural Adaptation, Validity, and Reliability Properties When Used with Children and Adolescents"

_children, 2021, doi:10.3390/children8121184_

Round 1

Reviewer 1 Report

The authors have submitted a clearly written and well researched Arabic validation of the widely used Faces Pain Scale (Revised) for the measure of intensity of paediatric pain. It is appropriate to have a version not only in Arabic language but with right-to-left orientation of the faces. Readers, I expect, would be interested in seeing the Arabic scale. I note that it was Figure 1, but that was not available to me. Generally this scale is used for children 5 to 12 years, but is valid for older people (above 12 years), although the numerical pain scales are more widely used for older adolescents and adults, and visual analogue scales when high validity is sought. Testing the age range 8-12 years separately was important to the extent that ideally there should be reference to results in this age range. I hope this can be accommodated in the abstract within the specified word count.

Real life (current) pain assessments are preferred, but such testing has previously been done extensively. Thus, it is reasonable to accept imagined pain for the purpose of the Arabic scale validation. It is rare for researchers of pain intensity to evaluate the influences of previous pain experiences on the interpretation of the top anchor of the scale and that would have been of interest although beyond the specific scope of this study.

It would be helpful to clarify whether the currently experienced pain (8.6%) was used in the analysis. Readers unfamiliar with Lebanese Grades would be helped by addition of usual age range eg Grade 4 (usual age range).

Readers need reminding that intensity is but one dimension to assess, even in acute pain but especially in chronic pain contexts (eg Jaaniste, T., Noel, M., Yee, R.D., Bang J., Tan, A.C. & Champion, G.D. Why unidimensional pain measurement prevails in the pediatric acute pain context and what multidimensional self-report methods can offer. Children 2019; 6: 132.

The results are of particular significance and interest to Arabic speaking health professionals, especially those in hospitals, in pain medicine, and in nursing.  

The quality of the study is fine and scientifically sound. The English language is good. 

Author Response

Reviewer 1:

The authors have submitted a clearly written and well researched Arabic validation of the widely used Faces Pain Scale (Revised) for the measure of intensity of paediatric pain. It is appropriate to have a version not only in Arabic language but with right-to-left orientation of the faces. Readers, I expect, would be interested in seeing the Arabic scale. I note that it was Figure 1, but that was not available to me. Generally this scale is used for children 5 to 12 years, but is valid for older people (above 12 years), although the numerical pain scales are more widely used for older adolescents and adults, and visual analogue scales when high validity is sought.

Authors’ response:  We thank the reviewer for the kind comments. We decided not to show the FPS-R as a figure as it is very well known. Inadvertently, we left a note, but it should have not been there. We have deleted the note about the figure.

Testing the age range 8-12 years separately was important to the extent that ideally there should be reference to results in this age range. I hope this can be accommodated in the abstract within the specified word count.

Authors’ response:  Following this suggestion, we have edited the Abstract, to explain that we used two situations based on the participants’ age. However it is not possible to add the results for the two groups. The maximum number of words is 200, and there are 199 words already.

Real life (current) pain assessments are preferred, but such testing has previously been done extensively. Thus, it is reasonable to accept imagined pain for the purpose of the Arabic scale validation. It is rare for researchers of pain intensity to evaluate the influences of previous pain experiences on the interpretation of the top anchor of the scale and that would have been of interest although beyond the specific scope of this study.

Authors’ response:  We concur with the reviewer: when possible, real life pain assessment is preferred. We also agree that evaluating the influences of previous pain experiences on the interpretation of the top anchor of the scale would have been beyond the scope of this study. We have added this as a suggestion for future research (see page 5).

It would be helpful to clarify whether the currently experienced pain (8.6%) was used in the analysis.

Authors’ response:  Participants’ pain intensity was not used in the analysis, and was only provided as descriptive information. Therefore, in order to avoid potential misunderstandings we have chosen to delete this from the revised manuscript. Table 1 has been deleted.

Readers unfamiliar with Lebanese Grades would be helped by addition of usual age range eg Grade 4 (usual age range).

Authors’ response:  As mentioned in our previous response, we have deleted Table 1 to avoid misunderstandings. The information about the participants’ age is more important than the grades. 

Readers need reminding that intensity is but one dimension to assess, even in acute pain but especially in chronic pain contexts (eg Jaaniste, T., Noel, M., Yee, R.D., Bang J., Tan, A.C. & Champion, G.D. Why unidimensional pain measurement prevails in the pediatric acute pain context and what multidimensional self-report methods can offer. Children 2019; 6: 132.

Authors’ response:  We concur with the reviewer. Pain intensity is just one domain, and should never be used to understand pain experience in children alone. However, the objective of our research was to study an adaptation of the FPS-R. We did not see how, or where, we could add this idea in the text. Therefore, we prefer to not elaborate on this particular issue this time.

The results are of particular significance and interest to Arabic speaking health professionals, especially those in hospitals, in pain medicine, and in nursing.  

The quality of the study is fine and scientifically sound. The English language is good. 

Authors’ response:  We thank the reviewer for the kind comments.

Reviewer 2 Report

Thank you for the opportunity to review this important and well-written piece. The manuscript titled, "The Arabic Version the Faces Pain Scale-Revised…,” makes an interesting contribution to general pediatrics and clinical psychology practice. As such, it is an excellent fit with the scope and mission of Children. Authors make a strong case for the value of the submitted study via their stated point of view that “it should not be assumed that the strong psychometric properties [of the FPS-R] reported in studies of samples of countries with different languages and cultures remains the same when it is used Arabic-speaking samples.” Thus, their effort is an appreciated one.

Nonetheless, this reviewer has identified a few areas of concern (from major to minor) that warrant reflection and consideration prior to an editorial decision.

1) “Imagine” pain (Lines 141 & 212) – authors describe a procedure in which participants imagined pain (a scraped knee or a burn) and subsequently rated that theorized experience. While there is a precedent noted, it is the opinion of this reviewer that precedent, alone, is insufficient to support the methodology. Authors acknowledge in the Discussion, albeit very briefly, that children may not have experienced burning their hand; however, the goes beyond this – even in cases where children have burned their hand, scraped their knees, etc., asking them to recollect the pain experience is far different than rating it in the moment. This reviewer recommends a more thorough discussion of this as a limitation of the study, as well as the proposal that, in future studies, this aspect of the methodology be improved.

2) Discriminant validity (Line 177) – authors suggest that appropriate discriminant validity was demonstrated by the fact that correlations of the FPS-R and the PCS-S were lower than those between the FPS-R and the NRS-11. Importantly, however, all of the correlations were statistically significant indicative of a meaningful relationship between the two (as opposed to a lack of relationship more suggestive of distinct constructs). This reviewer urges authors to acknowledge the variety of ways in which to assess discriminant validity and to temper their language about their findings accordingly. Please see: Mikko Rönkkö & Eunseong Cho. An Updated Guideline for Assessing Discriminant Validity. Organizational Research Methods; 2020. https://doi.org/10.1177/1094428120968614

3 Pain in participants (Lines 83) – authors note that a portion of their participants reported pain at the time of data collection. This reviewer asks that authors provide additional detail on this portion of the sample, including whether the pain was acute or chronic, present at the exact time of data collection, etc., as these factors are quite relevant in determining the utility of the FPS-R in various pain samples. Furthermore, please reconcile the different number of those with pain at the time of data collection (n=14 vs n=25) as stated in the text and in Table 1, respectively.

4) Reliability (Line 183) – authors state that the test-retest reliability coefficients can be found in Table 2; however, they appear to be missing. Please remove reference to Table 2 from the text or add the coefficients to the table, as stated.

Author Response

Thank you for the opportunity to review this important and well-written piece. The manuscript titled, "The Arabic Version the Faces Pain Scale-Revised…,” makes an interesting contribution to general pediatrics and clinical psychology practice. As such, it is an excellent fit with the scope and mission of Children. Authors make a strong case for the value of the submitted study via their stated point of view that “it should not be assumed that the strong psychometric properties [of the FPS-R] reported in studies of samples of countries with different languages and cultures remains the same when it is used Arabic-speaking samples.” Thus, their effort is an appreciated one.

Authors’ response:  We thank the reviewer for the kind comments.

Nonetheless, this reviewer has identified a few areas of concern (from major to minor) that warrant reflection and consideration prior to an editorial decision.

  • “Imagine” pain (Lines 141 & 212) – authors describe a procedure in which participants imagined pain (a scraped knee or a burn) and subsequently rated that theorized experience. While there is a precedent noted, it is the opinion of this reviewer that precedent, alone, is insufficient to support the methodology. Authors acknowledge in the Discussion, albeit very briefly, that children may not have experienced burning their hand; however, the goes beyond this – even in cases where children have burned their hand, scraped their knees, etc., asking them to recollect the pain experience is far different than rating it in the moment. This reviewer recommends a more thorough discussion of this as a limitation of the study, as well as the proposal that, in future studies, this aspect of the methodology be improved.

Authors’ response:  The reviewer has a good point here. Recollected pain intensity and actual pain intensity may not be equal. However, this is not important for the objectives of this study. Moreover, recalled pain intensity or imagined pain intensity reports are widely used in both clinical and experimental pain studies, and are well suited for the purpose of the Arabic scale validation. Experimentally induced pain would solve, in part, the problem raised here. However, although this procedure has been extensively used with adults, it is not as usual in children. Nevertheless, following this suggestion, we have edited the Discussion section. We describe the procedure used as a limitation to the study, and the need to improve the procedure in future studies by using samples of children with acute and chronic pain.

  • Discriminant validity (Line 177) – authors suggest that appropriate discriminant validity was demonstrated by the fact that correlations of the FPS-R and the PCS-S were lower than those between the FPS-R and the NRS-11. Importantly, however, all of the correlations were statistically significant indicative of a meaningful relationship between the two (as opposed to a lack of relationship more suggestive of distinct constructs). This reviewer urges authors to acknowledge the variety of ways in which to assess discriminant validity and to temper their language about their findings accordingly. Please see: Mikko Rönkkö & Eunseong Cho. An Updated Guideline for Assessing Discriminant Validity. Organizational Research Methods; 2020. https://doi.org/10.1177/1094428120968614

Authors’ response:  Again, the reviewer has a good point. There are different ways to study discriminant validity, as there are different ways to define this term. The fact that two constructs are related does not imply that they are measuring exactly the same construct. For example, pain catastrophizing has been repeatedly positively associated with pain intensity,  but research has shown that they are two constructs that are empirically distinguishable (which is one of the common definitions of discriminant validity). We used one of the different procedures that are available to study discriminant validity, one that has been repeatedly used with pain intensity questionnaires. The data showed that the scores have good, although not excellent, convergent and discriminant validity. Following this suggestion we have edited the text to address this issue and provide a more tempered position in the manuscript.    

  • Pain in participants (Lines 83) – authors note that a portion of their participants reported pain at the time of data collection. This reviewer asks that authors provide additional detail on this portion of the sample, including whether the pain was acute or chronic, present at the exact time of data collection, etc., as these factors are quite relevant in determining the utility of the FPS-R in various pain samples. Furthermore, please reconcile the different number of those with pain at the time of data collection (n=14 vs n=25) as stated in the text and in Table 1, respectively.

Authors’ response:  As we mentioned in a previous response to a comment from Reviewer 1, we have deleted this information from the manuscript. Current pain intensity data was not used in any way.

  • Reliability (Line 183) – authors state that the test-retest reliability coefficients can be found in Table 2; however, they appear to be missing. Please remove reference to Table 2 from the text or add the coefficients to the table, as stated.

Authors’ response:  We apologize for this mistake. Reference to Table 2 has been removed from the text.

Round 2

Reviewer 2 Report

To address this reviewer's #1 concern, authors added a statement to the Discussion that includes, "...and evaluate the influence of previous pain experiences on the interpretation of the top anchor of the questionnaire.”   The meaning of this statement is not clear to this reviewer, nor is it entirely clear how it addresses the reviewer's stated concern. 

2) Related to this reviewer's question related to discriminant validity , authors stated in their reply that they “edited the text to address this issue and provide a more tempered position in the manuscript.” However, this reviewer does not see that change. Can authors provide line numbers where the change occurred?

Author Response

1) To address this reviewer's #1 concern, authors added a statement to the Discussion that includes, "...and evaluate the influence of previous pain experiences on the interpretation of the top anchor of the questionnaire.”   The meaning of this statement is not clear to this reviewer, nor is it entirely clear how it addresses the reviewer's stated concern. 

Authors’ response: There is a misunderstanding here. To address the Reviewer #1 concern, we explained why we used the procedure that was implemented in the study. In addition, we also explained that the procedure used is very common in this type of studies. Specifically, we stated that “The reviewer has a good point here. Recollected pain intensity and actual pain intensity may not be equal. However, this is not important for the objectives of this study. Moreover, recalled pain intensity or imagined pain intensity reports are widely used in both clinical and experimental pain studies, and are well suited for the purpose of the Arabic scale validation. Experimentally induced pain would solve, in part, the problem raised here. However, although this procedure has been extensively used with adults, it is not as usual in children.”  

Moreover, we also stated that “Nevertheless, following this suggestion, we have edited the Discussion section. We describe the procedure used as a limitation to the study, and the need to improve the procedure in future studies by using samples of children with acute and chronic pain.”

The text that we added "...and evaluate the influence of previous pain experiences on the interpretation of the top anchor of the questionnaire.”  was related to a response to another reviewer and had nothing to do with this Reviewer’s comment. This addition to the text was included as an area of future research. Top anchors used in the assessment of pain intensity in children is an area of concern. That is, it is unclear the influence of the words used in the top anchor in the child’s pain intensity report. Although this was not of relevance to our study (it goes beyond the objectives of the validation study as it was also described by the reviewer), we wanted to rise it as an area of interest for future research.  

2) Related to this reviewer's question related to discriminant validity , authors stated in their reply that they “edited the text to address this issue and provide a more tempered position in the manuscript.” However, this reviewer does not see that change. Can authors provide line numbers where the change occurred?

Authors’ response: In our response to this Reviewer’s question we responded the following: “Again, the reviewer has a good point. There are different ways to study discriminant validity, as there are different ways to define this term. The fact that two constructs are related does not imply that they are measuring exactly the same construct. For example, pain catastrophizing has been repeatedly positively associated with pain intensity, but research has shown that they are two constructs that are empirically distinguishable (which is one of the common definitions of discriminant validity). [In this study] We used one of the different procedures that are available to study discriminant validity, one that has been repeatedly used with pain intensity questionnaires.

We also stated the following: “The data showed that the scores have good, although not excellent, convergent and discriminant validity. Following this suggestion we have edited the text to address this issue and provide a more tempered position in the manuscript.

Therefore, in this manuscript we emphasized that the discriminant validity is good (not excellent), which conveys the idea that discriminant validity is not perfect.

In addition, in trying to correspond to the suggestion of this reviewer, we went through the manuscript and edited what we understood was needed. In doing so, we deleted the statement written at the end of the section “3.3.2. Discriminant Validity” where we informed about the results related to the analysis of the discriminant validity. We cannot provide the number of the lines as we do not have access to the document that is used by the reviewers. The text that was deleted is the following “which supports the discriminant validity of the FPS-R scores.” We understand that this was hard to identify as it was a deletion not an addition or change, which would have been identified by our using a red font.